# Interactions between Gender and Sepsis—Implications for the Future

**DOI:** 10.3390/microorganisms11030746

**Published:** 2023-03-14

**Authors:** Ines Lakbar, Sharon Einav, Nathalie Lalevée, Ignacio Martin-Loeches, Bruno Pastene, Marc Leone

**Affiliations:** 1Department of Anesthesiology and Intensive Care Unit, Assistance Publique Hôpitaux Universitaires de Marseille, Aix-Marseille University, Hospital Nord, 13015 Marseille, France; 2CEReSS, Health Service Research and Quality of Life Centre, School of Medicine-La Timone Medical, Aix-Marseille University, 13015 Marseille, France; 3Intensive Care Unit, Shaare Zedek Medical Center, Jerusalem 23456, Israel; 4Faculty of Medicine, Hebrew University, Jerusalem 23456, Israel; 5INSERM, INRAE, Centre for Nutrition and Cardiovascular Disease (C2VN), Aix-Marseille University, 13005 Marseille, France; 6Intensive Care Unit, Trinity Centre for Health Science HRB-Wellcome Trust, St James’s Hospital, D08 NHY1 Dublin, Ireland

**Keywords:** gender, sex, sepsis, shock, infection

## Abstract

Sex and gender dimorphisms are found in a large variety of diseases, including sepsis and septic shock which are more prevalent in men than in women. Animal models show that the host response to pathogens differs in females and males. This difference is partially explained by sex polarization of the intracellular pathways responding to pathogen–cell receptor interactions. Sex hormones seem to be responsible for this polarization, although other factors, such as chromosomal effects, have yet to be investigated. In brief, females are less susceptible to sepsis and seem to recover more effectively than males. Clinical observations produce more nuanced findings, but men consistently have a higher incidence of sepsis, and some reports also claim higher mortality rates. However, variables other than hormonal differences complicate the interaction between sex and sepsis, including comorbidities as well as social and cultural differences between men and women. Conflicting data have also been reported regarding sepsis-attributable mortality rates among pregnant women, compared with non-pregnant females. We believe that unraveling sex differences in the host response to sepsis and its treatment could be the first step in personalized, phenotype-based management of patients with sepsis and septic shock.

## 1. Introduction

The terms gender and sex are often used interchangeably when discussing disease differences between males and females. However, gender refers to the social representation of a group while sex relates to biological characteristics. Gender-related disparities may be addressed, at least in part, by changing the socially constructed roles and behaviors of a population. A typical example is the decreasing rate of sexually transmitted diseases in populations at risk. Addressing sex-related differences requires in-depth epidemiological and biological understanding of disease processes.

Sex-based differences in disease are widely reported in the literature. For example, females have a higher prevalence of autoimmune disease [1], whereas males have a higher prevalence of cancer [2]. Sepsis, defined as life-threatening organ dysfunction resulting from a host’s dysregulated response to infection [3,4], is more prevalent among males [5,6,7]. The difference in sepsis prevalence stems from both social and biological causes. Social aspects include, for example, greater exposure to risk factors related to lifestyle (e.g., physical activity, work outdoors) and greater exposure to violence among men. Biological aspects relate to sex-related differences in hormonal and immune system characteristics. SARS-CoV-2 is one typical example, wherein both the incidence and the severity of COVID-19 disease among males are higher [8]. Women have a higher lifetime risk and worse disease expression of specific infections, such as tuberculosis and malaria [9]. Sex differences in hormonal and immune profiles are most extreme during pregnancy and the peripuerum. In the developing world, during this period one in five women contract a clinically significant infection [10]. This review highlights some of the more important experimental and clinical studies showing sex-specific differences in sepsis.

## 2. Animal Models

Multiple animal models have been established to investigate sepsis, including systemic administration of endotoxins or pathogens by various routes and cecal ligature and puncture. Animal models would enable assessment of the interaction between sex and sepsis while minimizing clinical confounders. However, the vast majority of pre-clinical sepsis studies have been performed only on male animals. The historic justification for the use of male animals only was the need to control for hormonal effects. However, early assumptions that female animals are prone to greater physiological variability due to hormonal cycles have not been supported by evidence [11,12,13,14]. Since it is becoming increasingly clear that female rodents are not small male rodents [15], this issue may render most basic research irrelevant for women. Research on female animal models is currently lagging at least 20–30 years following studies conducted on male animal models [16]. Therefore, with regards to sepsis and sex—what have we learned from animal models thus far?

### 2.1. Sex and the Immune System Response to Sepsis

Increasing evidence suggests that sex affects the host response to sepsis; sex hormones have been shown to target most immune system cells. For example, using a transcriptomic approach, Textoris et al. showed that 86% of genes involved in the response to *Coxiella burnetii* infection were associated with sex [17]. Such encompassing involvement suggests a very high likelihood of sex polarization in the host response to infection.

Overall, female sex seems protective with regards to the likelihood of sepsis; after an infectious challenge, bacteria are cleared faster in females than in males [17], whereas ovariectomized animals tend to have a bacterial clearance similar to that of male animals [17]. Females also seem to have a less reactive inflammatory response than males [17]. Estrogens induce efficient cell-mediated and humoral immune responses while androgens are suppressive [18]. The lower immune reactivity among females compared with males is believed to stem from evolutionary adaptations to pregnancy [19].

Multiple pathways lead to the difference in immune response between sexes. For example, dual-specificity phosphatase 3 (DUSP3) is linked to estrogen-mediated modulation of the macrophage response, thereby protecting female rats from endotoxemia-induced and polymicrobial-induced septic shock [20]. AMP-activated protein kinase (AMPK) isoform α1 (an enzyme with hepatoprotective effects during sepsis) modulates the susceptibility to multiple organ injury in a sex-dependent manner via its metabolic functions [21]. Adaptative immunity is also sex-dependent; estradiol boosts reactive oxygen species and promotes phagocytosis in murine sepsis models [22].

Finally, there is a close interaction between steroid receptors and sex. In fact, sex hormone receptors are also called sex steroid receptors, and they interact with sex hormones, including androgens, estrogen, and progesterone. The use of steroids in clinical practice (e.g., during septic shock) may interfere with sex hormone metabolism. While this topic has been partially explored [23], none of the studies conducted in animal models have taken into consideration confounding by external steroid use, a medication commonly used in septic shock. Clearly this requires further research [24].

### 2.2. Sex and the Cardiac Response to Sepsis

Several studies have reported fundamental data regarding the cardiac response to sepsis and sex dimorphism [25,26,27]. Estrogen modulates several acute injury-related myocardial responses. Endotoxin administration produces less pronounced myocardial dysfunction in females than in males [28]. Beta-blocker administration has been suggested to be beneficial for patients with septic shock as it may improve the cardiac relaxation, left ventricular filling, and the cardiac index [29]. However, sex differences have been shown in the beta-adrenergic response of isolated rabbit hearts [30]. In an in vivo model of landiolol (a short acting beta blocker) administered after cecal ligation and puncture in rats, cardiac function worsened significantly after CLP and landiolol reversed this effect in males. Conversely, landiolol decreased left ventricle ejection fraction in females [31]. A subsequent transcriptome study showed that landiolol reverses the expression of several genes that are downregulated during sepsis in male rats. However, sepsis-induced downregulation of gene expression was less pronounced in females than in males, and was maintained in landiolol-treated female rats [32]. A third study randomized female rats to ovariectomy and showed that after CLP, cardiac performance decreased in the ovariectomized female rats. Inflammatory and apoptotic pathways, including the JAK/STAT pathway, were less activated in ovariectomized septic females than in controls, as well as adrenergic and calcium pathways, in line with decreased cardiac performance and poor hemodynamic profiles. Conversely, administration of landiolol increased calcium entry into the cardiomyocytes of ovariectomized female rats. Landiolol restored JAK2 expression and overexpressed genes responsible for calcium influx in ovariectomized septic females, whereas it did the opposite in control females [33]. This may explain why cardiac performance improves only in ovariectomized septic females (i.e., less exposure to female hormones during sepsis).

At the cellular level, there are sex-related differences in the inflammatory response; calcium signaling and apoptosis pathways and a protective cardiac effect in female animals have been associated with increased AKT and endothelial NO synthase (eNOS) phosphorylation, which may be involved in the response to beta blockers [28,31]. Overexpression of the eNOS protein [31] and decreased level of the guanylate cyclase 1 soluble subunit alpha 2 (GUCY1A2) transcript [32], regulators of cGMP, were found only in septic males. In male rats, landiolol reversed the increase in eNOS expression, whereas no such effect was observed after landiolol administration in females [32]. Cyclic guanosine monophosphate (cGMP) is protective in males and deleterious in females [27]. Sex-related differences in regulation of β-adrenergic signaling and calcium cycling were associated with failure of diastolic relaxation and impaired systolic contraction [34], decreased stroke volume index [31], and reduced survival in septic males [35]. A decrease in myofilament calcium sensitivity following troponin I hyperphosphorylation in males may explain this difference [35].

Transcriptional research also highlights male-specific regulation of the JAK/STAT, phosphoinositide-3-kinase (PI3K), and focal adhesion pathways as well as overexpression of p53-dependent cell-cycle arrest (p21 and stromal antigen 1 (STAG1)), toll-like receptor 1 (TLR1), and the myeloid differentiation primary response 88 (MyD88) [32]. These changes, not described in females, have been associated with a poor outcome [35]. Moreover, estrogen may modulate ryanodine receptor properties [36] and β-adrenergic receptors to affect Ca2+-handling proteins and the PI3K-AKT pathway [37]. Sepsis induced cardiac dysfunction improves after administration of 17-β-œstradiol [38] and immunomodulatory effect of estrogens in animal models [39] (Figure 1).

### 2.3. Sex and the Microbiome in Sepsis

The microbiome has recently become a topic of interest among several mechanisms of the host response to sepsis [40]. Microbiome diversity decreases in both male and female mice after cecal ligation and puncture. However, at day 14, the microbiome of females recovers fully while that of males still differs from controls. Notably, female recovery seems age dependent; only young females are similar to controls at day 14 [41]. The authors that showed these findings have suggested that female resilience to microbiome pathogens may explain some of the sex-related differences in sepsis outcomes.

### 2.4. Sex and the Central Nervous System in Sepsis

A murine model of the central nervous system response to sepsis found that sex and age influenced both the response of the brain to sepsis and its recovery after sepsis [42]. Aging was associated with delayed recovery from sepsis, and this delay was more pronounced in males. In a rat model of septic shock, transcriptional responses linked to neurogenesis and myelination were downregulated in all rats, but recovery rates differed by sex and age. Young female rats had the most rapid recovery rate. In older female rats, although most genes recovered, those linked to neurogenesis and myelination were still downregulated but no longer decreasing on day 4. In older male rats, genes linked to neurogenesis and myelination continued to decrease on day 4, while immune response genes continued to increase. A subsequent epigenetic analysis showed that on days 1 to 4 after sepsis, young female rats had better brain microRNA response profiles than young males and older females [43].

### 2.5. Pregnancy and Sepsis

The maternal immune system adapts during pregnancy to avoid allogenic rejection of the fetus, which is genetically distinct [44]. Most animal models of sepsis in pregnant animals have focused on fetal effects and neonatal outcomes [45]. We are unaware of animal studies specifically intended to address the effects of maternal immune system adaptations to pregnancy on maternal effects and outcomes in sepsis.

### 2.6. Summary of Lessons from Animal Models of Sepsis and Sex

Many papers have highlighted the role of sex dimorphism on the host response to sepsis. In brief, females are less susceptible to sepsis and seem to recover more effectively than males. Age remains an important confounder that requires further investigation; old females clearly differ from young females with regards to the host response to sepsis. While this may seem entirely attributable to changes in sex hormones, interference by other variables has not been ruled out. Among the studies conducted on interventions during sepsis (e.g., landiolol, antimicrobials, fluids), all have shown some evidence of a difference between sexes when such a difference was sought [46], but these findings are often contradictory [47,48].

Of note, animal models have several limitations with regards to the topic at hand. First, no model fully reflects the complexity and the varied presentation of sepsis. Careful planning is implemented when designing research protocols for sepsis in animal models; as physiological variability decreases, result interpretation is more precise [49]. Second, the effects of prior diseases and/or comorbidities are not really addressed in most models. Finally, most animal models of sepsis do not take into account organ support which could be an important source of confounding [50].

## 3. Clinical Studies

Twenty years after the European Union policy declaration intended to encourage and improve sex and gender aspects of biomedical and health-related research [51], the critical care community remains reluctant to address topics relating to differences between men and women in disease epidemiology, manifestations, and outcomes. As it is important to understand the role of sex in the epidemiology and clinical presentation of sepsis [52], the following section will address the few advances that have been made in understanding sex differences in sepsis among human subjects.

### 3.1. Sex Differences in Sepsis Epidemiology

Several studies have reported sex-based differences in the epidemiology of sepsis and all of these studies show a consistently higher risk of sepsis in men (Table 1). One nationwide study reported that male patients contributed 63.8% of the 187,587 septic shock episodes included in the study [53]. Even during the steady increase in the incidence of septic shock observed during 4 years, this sex discrepancy remains consistent [53].

The reasons underlying the higher rate of sepsis in men remain unclear. Several explanations have been put forward for this phenomenon, with the truth probably being a combination of all of those. Hypotheses include physiological characteristics affecting different susceptibility to infection, a greater inclination to progress from non-severe to severe infection, and gender differences in sepsis management. Additional factors may also play a role in the interaction between sepsis and sex. Smoking, alcohol consumption, and recreational activities differ between men and women. Such causes may also affect the likelihood of environmental exposure to infectious diseases (more specifically pathogens) [54,55].

### 3.2. Sex Differences in the Sources of Infection

Men are more commonly affected by endocarditis and bacteremia [56,57,58]. Men also tend to develop sepsis more frequently from respiratory infections (36% vs. 29% for women, *p* < 0.01), while women tend to develop sepsis more frequently from genitourinary infections (35% vs. 27% for men, *p* < 0.01) [7].

Accordingly, different pathogens are more commonly identified as causative of sepsis in men and women. For instance, bacteremia caused by *Staphylococcus aureus* and *Pseudomonas aeruginosa* is more prevalent in males, with a male-to-female ratio of 1.5 and 2.8, respectively [59,60]. Conversely, 60% of *Escherichia coli* bacteremias occur in females, which is consistent with the higher incidence of urinary tract infections in females [61]. Men have a higher rate of sepsis caused by *Candida* species; the male-to-female ratio of candidemia ranges from 1.16 to 1.65 in various reports [62,63,64].

### 3.3. Sex Differences in Organ Dysfunction and Treatment

A study of 18,757 patients reported that the management of sepsis was more conservative in women than in men [65,66,67]. Women were less likely to receive hemodialysis catheters (OR = 0.85; 95% CI, 0.78–0.93) [65], deep venous thrombosis prophylaxis (OR = 0.90; 95% CI, 0.84–0.97), and invasive mechanical ventilation (OR = 0.81; 95% CI, 0.76–0.86) [65].

No difference has been reported in the incidence of sepsis-associated acute kidney injury in men and women [68]. However, a meta-analysis of 21 studies including a total of 545,538 participants revealed that females were less likely to receive renal replacement therapy compared with males (adjusted OR 0.81 [0.73–0.89], I^2^ = 57.4%) [69].

The reported rates of sepsis-associated myocardial dysfunction range from 10% to 70% [70]. Substantial differences exist between men and women in the prevalence, manifestations and clinical presentation, and outcomes of heart disease [71,72], yet human data on this topic in relation to sepsis remains scarce. Male sex has been associated with a higher incidence of myocardial dysfunction, compared with females, in both sepsis and septic shock [73].

The administration of steroids interferes with the natural functions of endogenous sex hormones. In the ADRENAL trial (n = 3713 patients), differences were identified between men and women in the effects of hydrocortisone during sepsis: hydrocortisone increased the risk of shock recurrence in females (OR 1.48 [1.03–2.14], *p* = 0.03) [74].

### 3.4. Sex Differences in Sepsis Outcomes

The literature examining the relationship between sex and sepsis-associated mortality is inconclusive. Some studies report higher mortality rates among men with sepsis [75,76,77,78] and other studies report higher mortality rates among women with sepsis [58,65,79,80,81,82] (Table 1). However, several studies also found no statistically significant difference in the mortality rates of men and women with sepsis [6,56,83,84,85]. A systematic review and meta-analysis of 13 studies (80,520 participants) published between 2007 and 2020 also found no significant differences in all-cause hospital and ICU mortality between men and women with sepsis, but the level of evidence for this finding was very low [86]. Furthermore, meta-analysis identified higher 28-day all-cause mortality and lower 1-year all-cause mortality among women. This evidence was also classified as being of very low-certainty and low-certainty, respectively [86].

Several reasons may underlie these conflicting findings. Comorbidities may play an important role in patient outcomes [87], but few epidemiological studies adjust for these variables. As noted elsewhere, silent variables are always a potential source of bias. The larger the dataset and the smaller the number of variables adjusted for—the larger the potential effect of such bias if it does exist [88]. A typical example of a comorbidity that should be adjusted for is chronic obstructive respiratory disease, which is more prevalent among men than among women [87]. Men also display heart disease at a younger age than women [87]. In one prospective cohort which included 64,040 participants, the risk of bloodstream infection was reported to be 41% higher among men than among women and one-third of this excess risk was mediated by cardiovascular risk factors and comorbidities [89].

The rates of withholding/withdrawing of care are also rarely reported in sepsis studies. Code status limitations are ordered more often for women than for men (OR = 1.31; 95% CI, 1.18–1.47) [65]. However, while decisions to withhold or withdraw potentially life-prolonging treatment were more often made in women (28.0% vs. 22.8%, *p* = 0.003), these decisions may actually be based on the different disease and comorbidity profiles of men and women [90].

Additional variables that may have led to at least some of the discrepancies observed in outcomes of men and women with sepsis are the severity of sepsis, which is often not taken into consideration in retrospective and database studies [91], patient frailty [92], and the rates of organ support provided. Few, if any, studies, adjust for these important confounders, and none has taken all of these into consideration.

Finally, studies examining the relationship between sex and sepsis outcomes may have included selective populations (e.g., dependent on the characteristics of the population served or the number of available ICU beds) [93]. Different study methods (e.g., prospective versus retrospective or database studies, single versus multicenter) could also account for the variability in outcomes.

**Table 1 microorganisms-11-00746-t001:** Details of studies reporting outcomes in sepsis and septic shock patients with respect to sex.

Study	Location	Years	Population	Number of Patients	Definition of Sepsis	Main Outcome	Incidence	Type of Mortality	Crude Mortality		Adjusted OR Mortality	Conclusions
[6]	Nationwide, U.S.	1979–2000	Sepsis patients	10,319,418 sepsis patients	ICD-9-CM codes	Epidemiologic characterization of sepsis	Males > females	In-hospital mortality	22% males	21.8% females	Not provided	No significant difference
[79]	Charité Hospital, Berlin	2006–2007	ICU patients, subgroup of sepsis	327 sepsis patients	SIRS + infection	ICU mortality rates for male and female patients	Males > females	ICU mortality	13.7% males	23.1% females	(reference male) 1.909 [1.002–3.638], *p* = 0.05	Higher mortality in females
[80]	24 ICUs in Italy	2006	Severe sepsis	200 severe sepsis/septic shock patients	ACCP/SCCM Consensus Conference Committee definition	ICU mortality rates for male and female patients	Males > females	ICU mortality	46.4% males	63.5% females	(reference male) 2.33 [1.23–4.39], *p* = 0.010	Higher mortality in females
[58]	Single centre, France	1995–2004	Severe nosocomial infections	1341 severe nosocomial infections	Definition of each nosocomial infection	ICU mortality rates for male and female patients	Males > females	ICU mortality	32% males	37% females	(reference male) 1.50 [1.11–2.03], *p* = 0.008	Higher mortality in females
[77]	Outcome rea database (12 French ICUs)	1997–2005	Severe community-acquired sepsis	1608 severe sepsis patients	SIRS + one (or more) organ dysfunction	ICU mortality rates for male and female patients	Males > females	ICU mortality	29% males	26% females	(reference male) 0.75 [0.58–0.98], *p* = 0.03	Lower mortality in females
[84]	Single centre, Germany	1995–2000	Patients with severe infection hospitalized in the ICU	308 severe infections	Definition of each infection	Hospital mortality	No difference	In-hospital mortality	31.7% males	34% females	Not provided	No significant difference
[75]	Single centre, Germany	Unknown	Severe sepsis and septic shock	11 severe sepsis, 41 septic shock	Definition of Bone et al. (1992)	Hospital mortality	Males > females	In-hospital mortality	70% males	26% females	Not provided	Lower mortality in females
[83]	State database, U.S. (California)	2005–2010	Severe sepsis	1,213,219 severe sepsis	Definition of Angus et al. (2001)	Identification of patient demographic, patient health	Females > males	In-hospital mortality	17.9% males	16.6% females	(reference female) 1 [1-1], *p* < 0.001	No significant difference
[81]	Single centre, U.S.	1994–1998	Sepsis	1348 sepsis patients	SIRS + infection	Mortality, ICU LOS, hospital LOS, and maximal multiple organ dysfunction score	Females > males	In-hospital mortality	21.4% males	25.3% females	*p* = 0.02 (F-Test, multivariate analysis)	Higher mortality in females
[85]	Single centre, U.S.	2005–2012	Severe sepsis or septic shock	814 severe sepsis or septic shock patients	Severe sepsis: 2 SIRS criteria + suspected infection + at least 1 organ dysfunction. Septic shock: the above + hypotension after a fluid bolus.	Completion of the SSC resuscitation bundle as previously defined by the SSC	Males > females	In-hospital mortality	23.4% males	25.2% females	Not provided	No significant difference
[74]	Nationwide, Australia	2006–2009	Severe sepsis and septic shock	12,912 sepsis patients	ICD 10th Revision, Australian modification codes	Mortality, ICU LOS, hospital LOS, and readmissions	Males > females	90-day mortality	25.3% males	22.5% females	Not provided	Lower mortality in females
[82]	Nationwide, Sweden	2008–2015	Severe sepsis and septic shock	2720 severe sepsis and septic shock patients	1992 sepsis definition	Management and outcome of sepsis patients	Males > females	30-day mortality	23.1% males	25% females	(reference male) 1.28 [1.00–1.64]	Higher mortality in females
[65]	98 ICUs, U.S., Canada, Brazil	2003–2006	Severe sepsis and septic shock	18,757 severe sepsis or septic shock	One severe acute organ dysfunction within 3 days of a presumed infection	Hospital mortality	Males > females	In-hospital mortality	33% males	35% females	(reference male) 1.11 [1.04–1.19]	Higher mortality in females
[76]	Single centre, U.S.	2001–2012	Severe sepsis and septic shock	6134 sepsis patients	Not detailed	1 year mortality	Males > females	1-year mortality	55.6% males	51.4% females	(reference female) 1.08 [1.00–1.16], *p* = 0.03	Lower mortality in females
[56]	2 ICUs, Netherlands	2011–2014	Sepsis	1815 sepsis patients	2001 International Sepsis Definitions	Differences in sepsis presentation and long-term outcome	Males > females	90-day mortality	45.5% males	43.4% females	(reference female) 1.06 [0.83–1.35]	No significant difference

Abbreviations: OR: odds ratio; ICU: intensive care unit; LOS: length of stay; ICD: international classification of diseases.

### 3.5. Pregnancy

Sepsis is ranked as the third leading cause of death among pregnant females, accounting for approximately 11% of maternal deaths and an 8% case fatality rate [94,95]. Physiological changes during pregnancy mirror those of sepsis (i.e., higher heart and respiratory rates, increased neutrophil counts, and decreased platelet counts), thereby rendering diagnosis more challenging [96]. Furthermore, since the fetus is not classified as an “organ”, in the absence of organ dysfunction, the occurrence of miscarriage, stillbirth, and neonatal complications secondary to maternal infection are not considered maternal complications of sepsis.

Conflicting data have been reported regarding sepsis-attributable mortality rates among pregnant women, compared with non-pregnant females [94,95]. Pregnant women are particularly susceptible to viral infections and the severity of the latter is increased compared with non-pregnant females. For instance, the likelihood of death from influenza is twice higher among pregnant women than among non-pregnant women of similar age. Pregnant women have also been reported to have higher mortality rates from COVID-19 infection than their age-matched counterparts [97,98].

However, a retrospective cohort analysis of 5968 pregnant and 85,240 non-pregnant females showed that sepsis-attributable case fatality rates remained higher in non-pregnant women than in pregnant women after adjustment for socio-economic status, race, age, and chronic comorbidities, which led the authors to conclude that pregnancy is associated with a protective effect which reduces the severity of sepsis presentation [99].

Given the significant hormonal changes that occur during pregnancy, it may be inferred that hormones play a significant role in the outcome of sepsis in these cases; however, this needs further elucidation.

### 3.6. Sex Differences Related to Culture

The recent pandemic has highlighted the glaring ethical implications of excluding pregnant and lactating women and women of childbearing age from clinical trials [100,101,102,103]. To date, most large randomized controlled trials have excluded women of childbearing age and pregnant women are still being excluded from trials designed by intensivists [104]. Continuing this practice perpetuates the lack of data and potential treatment inequities.

The medical community has already recognized the male advantage in rapid recognition of heart disease and rapid treatment [105]. The presence of similar bias has been suggested for sepsis. One study evaluated timely identification of sepsis using a clinical vignette in 120 physicians. Respondents identified sepsis with greater accuracy in women than in men [106]. Contrastingly, in the department of emergency medicine, the likelihood of progression from sepsis to septic shock has been found to be six times higher among women than among men [107], suggesting late identification and/or treatment. Indeed, registry data have shown that women treated by emergency medical service and emergency department staff were less likely to receive intravenous fluid and oxygen among men and that men receive antibiotics more rapidly than women (65 min (IQR 30-136) vs 87 min (IQR 39-172) (*p* = 0.0001)) [82].

Of even greater concern is the possibility that outcomes of men and women may differ according to the sex of the treating clinician. A Canadian database study examining 21 surgical procedures (n = 1,320,108 patients) found worse outcomes among women patients treated by male surgeons [108]. This finding was not supported in a study of patient outcomes after out-of-hospital cardiac arrest treated by male or female physicians [109].

Finally, physiological differences in the response to treatment may also affect outcome despite the best performance and intentions.

### 3.7. Perspectives

In terms of future strategies, the cornerstone of these findings should be first the principle of equity in care. As previously mentioned, studies highlighted disparities in treatment between male and female patients, as well as differences in outcomes according to the gender of physicians. Consequently, it is imperative to dedicate efforts towards the education of young medical professionals, with a focus on raising awareness about implicit biases related to gender. This awareness will contribute to reduce the impact of such biases on clinical decision making, and ultimately promote equitable treatment and outcomes for all patients [67,110,111,112].

Furthermore, it is also critical to focus on pharmacokinetics as another key area for intervention. There are known differences in drug metabolism according to gender. For example, drugs that prolong the QT interval are more likely to cause lethal ventricular arrhythmias in females than males [113]. As mentioned above, the increased efficacy of beta blockers in male animals than in their female counterparts [31,32] raises questions regarding their indiscriminate use in the management of sepsis and septic shock in human patients. Experimental evidence on this topic remains limited, as underlined by the study of Zhang et al. who encountered methodological challenges while attempting to conduct a comprehensive meta-analysis on the effect of biological sex on treatment response to fluid and antibiotic therapy in animal models of sepsis [46]. This was due to the paucity of available literature with only two relevant articles being identified through their search process [46]. Therefore, it is important to integrate considerations of sex differences in pharmacokinetics into clinical practice, as well as further research into the development of more personalized and tailored treatments based on sex-related factors.

## 4. Conclusions

Many important physiological issues relating to the relation between sex and sepsis remain to be studied. The lifetime risk for heart failure with reduced left ventricular ejection fraction is much higher among men than among women [114]. Women tend to have heart failure with preserved ejection fraction (i.e., right heart failure) and are also more symptomatic than their male counterparts [115]. Whether this affects sepsis outcomes remains to be seen. Women tend to have more complications after cardiovascular interventions and have more medication side effects [116,117,118]. Whether this also pertains to medications and interventions during sepsis is unknown. Women with heart disease wait longer before seeking treatment following an acute myocardial infarction [105,119]. Perhaps the opposite is true of men with infection.

At this time, the quality of the existing evidence does not enable drawing any definitive conclusions regarding the association between sex and sepsis-associated mortality. In order to claim equity exists, it does not suffice to show similar outcomes. Risk factors and process-related variables must be studied in depth. For all we know, the mortality rates attributed to sepsis in either men or women may still be overly high due to easily correctible factors that remain unaddressed.

In males (top panel), sepsis induces severe cardiac dysfunction by activating pathways responsible for inflammation, apoptosis, and oxidative stress, and by inhibiting pathways associated with excitation/contraction coupling (indications in red). Landiolol, by acting on the regulation of several genes, strongly improves cardiac function by reversing these biological processes (blue indications).

In females (bottom panel), the effect of sepsis is markedly weaker than in males and the cardiac dysfunction is much more pronounced (indications in red). In contrast to the male, landiolol has little effect on gene regulation. It does not improve the effect of sepsis on biological processes such as inflammation or oxidative stress but, on the contrary, acts on the b-adrenergic pathway and further decreases the contraction and relaxation capacity of the heart (blue indications).

## Figures and Tables

**Figure 1 microorganisms-11-00746-f001:**
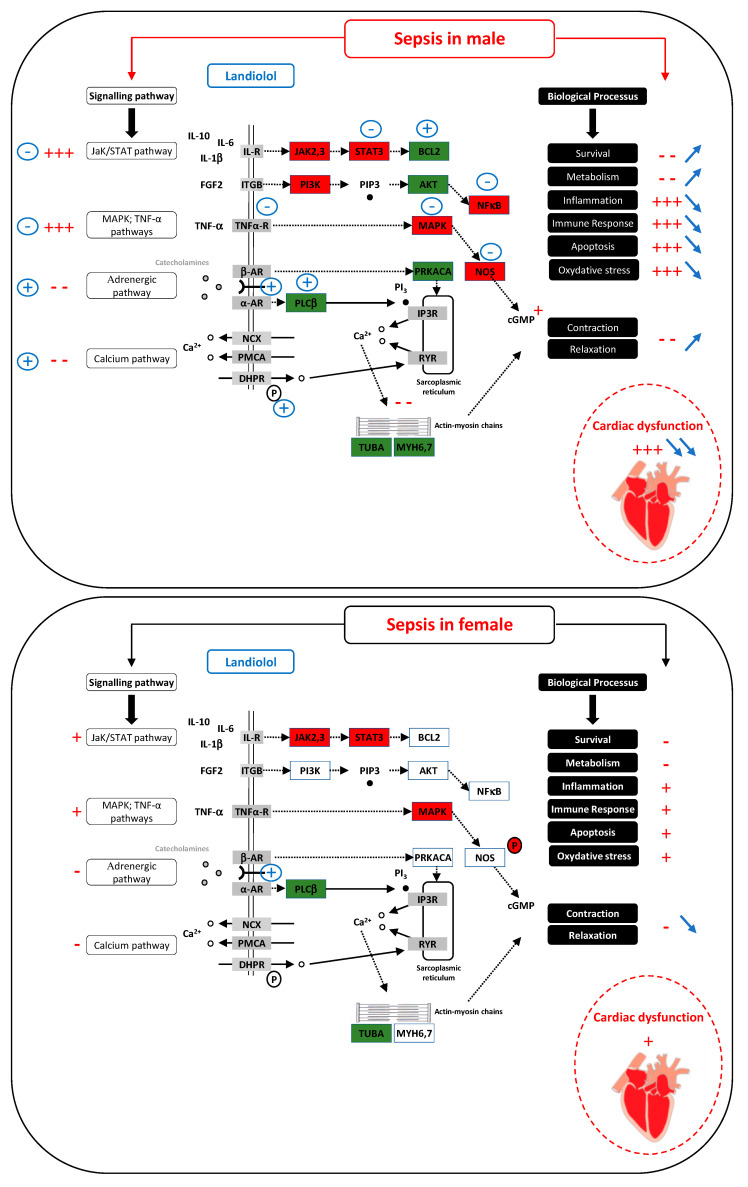
In males (top panel), sepsis induces severe cardiac dysfunction by activating pathways responsible for inflammation, apoptosis, oxidative stress and by inhibiting pathways associated with excitation/contraction coupling (indications in red). Landiolol, by acting on the regulation of numerous genes, strongly improves cardiac function by reversing these biological processes (blue indications). In females (bottom panel), the effect of sepsis is markedly weaker than in males and the cardiac dysfunction is much more pronounced (indications in red). In contrast to the male, landiolol has little effect on gene regulation. It does not improve the effect of sepsis on biological processes such as inflammation or oxidative stress but, on the contrary, acts on the beta-adrenergic pathway and further decreases the contraction and relaxation capacity of the heart (blue indications). Cellular pathways according to sex. + = activation; − = inhibition; 
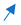
 = increase; 
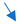
 = decrease; red rectangle = upregulated gene; green rectangle = downregulated gene.

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
