# Peer review of "Interactions between Gender and Sepsis—Implications for the Future"

_microorganisms, 2023, doi:10.3390/microorganisms11030746_

Round 1

Reviewer 1 Report

Sepsis constitutes an emerging state in medicine. Although a great improvement in its management has been achieved during the years, new possible strategies are still crucial to be found. Thus, going this path, the authors tried to highlight the involvement of sex in the course of sepsis’ treatment, referring to already conducted studies. Generally, the review is well written, but mainly based on already known information. I miss the paragraph focusing on possible new strategies of the sepsis’ treatment from the perspective of a different gender.

Author Response

Dear Reviewer 1,

It  is our pleasure to respond to each of your comments. We thank you for this relevant review that allowed us to improve our manuscript.

This review summarized the sex and gender cofactor in sepsis, a neglected area in this field. This review is comprehensive and well written and will cause more attention on this aspect in future studies.

We thank Reviewer 1 for this positive review of our manuscript. We are delighted to hear that you found our review to be comprehensive and well written, and we appreciate your comment that it will bring more attention to this important aspect in future studies. We hope that our work will contribute to a better understanding of the role that sex and gender play in sepsis.

Minor points:

The abstract requires some detailed points

We thank you for this comment. The abstract has been detailed as follows:

Sex- and gender-dimorphisms are found in a large variety of diseases, including sepsis and septic shock which are more prevalent in men than in women. Animal models show that the host response to pathogens differs in females and males. This difference is partially explained by sex-polarization of the intracellular pathways responding to pathogen-cell receptor interactions. Sex hormones seem to be responsible for this polarization, although other factors, such as chromosomal effects, have yet to be investigated. In brief, females are less susceptible to sepsis and seem to recover more effectively than males. Clinical observations produce more nuanced findings, but men consistently have an higher incidence of sepsis, and some reports also claim higher mortality rates. However, variables other than hormonal differences complicate the interaction between sex and sepsis, including comorbidities as well as social and cultural differences between men and women. Conflicting data has also been reported regarding sepsis attributable mortality rates among pregnant women, compared to non-pregnant females. We believe that unraveling sex differences in the host response to sepsis and its treatment could be the first step in personalized, phenotype-based management of patients with sepsis and septic shock.

The text in the figure are hardly read and need to improve. A brief legend is required.

The quality of the figure has been modified accordingly. The following legend has been added:

Figure 1. Cellular pathways according to sex.

In males (top panel), sepsis induces severe cardiac dysfunction by activating pathways responsible for inflammation, apoptosis, oxidative stress and by inhibiting pathways associated with excitation/contraction coupling (indications in red). Landiolol, by acting on the regulation of numerous genes, strongly improves cardiac function by reversing these biological processes (blue indications).

In females (bottom panel), the effect of sepsis is markedly weaker than in males and the cardiac dysfunction is much more pronounced (indications in red). In contrast to the male, landiolol has little effect on gene regulation. It does not improve the effect of sepsis on biological processes such as inflammation or oxidative stress but, on the contrary, acts on the b-adrenergic pathway and further decreases the contraction and relaxation capacity of the heart (blue indications).

+ = activation; - = inhibition;    = increase;    = decrease; red rectangle = up-regulated gene; green rectangle = down-regulated gene

We thank you once again for your review and hope we have answered all  queries and comments in a satisfactory manner

Sincerely yours

Ines Lakbar, Marc Leone, on behalf of co-authors

Reviewer 2 Report

This review summarized the sex and gender cofactor in sepsis, a neglected area in this field. This review is comprehensive and well written and will cause more attention on this aspect in future studies.

Minor points:

The abstract requires some detailed points

The text in the figure are hardly read and need to improve. A brief legend is required.

Author Response

Dear Reviewer 2,

Thank you for your constructive comments which have highlighted several important issues that required amendment and clarification. We hope that the revised version is clearer and meets expectations.

Sepsis constitutes an emerging state in medicine. Although a great improvement in its management has been achieved during the years, new possible strategies are still crucial to be found. Thus, going this path, the authors tried to highlight the involvement of sex in the course of sepsis’ treatment, referring to already conducted studies.

Generally, the review is well written, but mainly based on already known information. I miss the paragraph focusing on possible new strategies of the sepsis’ treatment from the perspective of a different gender

Thank you for your valuable comment and helpful suggestion on our manuscript. We appreciate your recognition of the importance of this area in sepsis research.

To address your concern regarding new strategies, the following paragraph has been added to the manuscript: 3.7 Perspectives

3.7 Perspectives

In terms of future strategies, the cornerstone of these findings should be first the principle of equity in care. As previously mentioned, studies highlighted disparities in treatment between male and female patients, as well as differences in outcomes according to the gender of physicians. Consequently, it is imperative to dedicate efforts towards the education of young medical professionals, with a focus on raising awareness about implicit biases related to gender. This awareness will contribute to reduce the impact of such biases on clinical decision-making, and ultimately promote equitable treatment and outcomes for all patients [67,110–112].

Furthermore, it is also critical to focus on pharmacokinetics as another key area for intervention. There are known differences in drug metabolism according to gender. For example, drugs that prolong the QT interval are more likely to cause lethal ventricular arrhythmias in females than males [113]. As mentioned above, the increased efficacy of beta-blockers in male animals than in their female counterparts [31,32] raises questions regarding their indiscriminate use in the management of sepsis and septic shock in human patients. Experimental evidence on this topic remains limited, as underlined by the study of Zhang et al, who encountered methodological challenges while attempting to conduct a comprehensive meta-analysis on the effect of biological sex on treatment response to fluid and antibiotic therapy in animal models of sepsis [46]. This was due to the paucity of available literature with only two relevant articles being identified through their search process [46]. Therefore, it is important to integrate considerations of sex differences in pharmacokinetics into clinical practice, as well as further research into the development of more personalized and tailored treatments based on sex-related factors.

We thank you for your time and attention, and we are grateful that you give us the opportunity to correct and improve our manuscript. We hope that these improvements will bring this manuscript to the level of your expectations.

Sincerely yours

Ines Lakbar, Marc Leone, on behalf of co-authors